# Time rescaling of nonadiabatic transitions

Takuya Hatomura[1*]

**1** NTT Basic Research Laboratories & NTT Research Center for Theoretical Quantum Physics,
NTT Corporation, Kanagawa 243-0198, Japan
* takuya.hatomura.ub@hco.ntt.co.jp

November 8, 2022

## 1 Abstract

**Applying time-dependent driving is a basic way of quantum control. Driven systems show various dynamics as its time scale is changed due to the different amount of nonadiabatic transitions. The fast-forward scaling theory enables us to observe slow (or fast) time-scale dynamics during moderate time by inducing additional driving. Here we discuss its application to nonadiabatic transitions. We derive mathematical expression of additional driving and also find a formula for calculating it. Moreover, we point out relation between the fast-forward scaling theory for nonadiabatic transitions and shortcuts to adiabaticity by counterdiabatic driving.**

## 1 Introduction

Realization of high-speed quantum control is one of the most critical elements of quantum technologies. As a matter of course, even classical technologies have been developed in pursuit of high-speed processing for practical use. However, there is more essential reason in the quantum case. In quantum systems, decoherence is inevitable and it smears quantumness.

Speedup of quantum control is required for minimizing a bad influence of decoherence. Preciseness of quantum control is also an important factor. Indeed, most quantum advantages stem from delicate interference among the exponentially large number of quantum states or high sensitivity of quantum states against small system parameters. To realize such precise quantum control, its speed might have to be slower than experimental limitations to some extent.

Time rescaling of control schemes may be necessary to satisfy the above requirements. However, changing time scale affects dynamics and its measurement outcomes since the amount of nonadiabatic transitions differs. This is also a problem from the viewpoint of quantum simulation of nonadiabatic phenomena. The fast-forward scaling theory was proposed as a candidate for resolving this problem [1, 2]. It enables us to change time scale of dynamics without changing measurement outcomes by inducing additional driving. It was first formulated for a single-particle problem in potential [1], but it is not limited to such a specific system. Indeed, it has been extended to charged particles [3], many-body systems [4], discrete systems [5, 6], Dirac dynamics [7], classical systems [8], stochastic systems [9], etc (see Ref. [2] and references therein).

The fast-forward scaling theory can also be applied to acceleration of adiabatic time evolution by introducing a "regularization term" [10]. In this sense, the fast-forward scaling theory is regarded as one of the methods of shortcuts to adiabaticity [11–14]. There are two representative approaches in shortcuts to adiabaticity. One is counterdiabatic driving, in which speedup of adiabatic time evolution is realized by applying additional driving (the counterdiabatic term) [11, 12]. The other is invariant-based inverse engineering, in which it is realized by scheduling system parameters [13]. Relation between the fast-forward scaling theory and invariant-based inverse engineering was discussed in a specific system [15]. Moreover, it was pointed out in Ref. [16] that the regularization term is identical to the counterdiabatic term (or the single-eigenstate counterdiabatic term proposed in Ref. [17]). Combination of the fast-forward scaling theory and shortcuts to adiabaticity was also discussed [6].

Here we summarize points to be discussed in the present paper. First, we consider application of the fast-forward scaling theory to nonadiabatic transitions. Although the fast-forward scaling theory was originally formulated for nonadiabatic dynamics, it rather represents "not adiabatic" dynamics. We formulate it so that nonadiabatic transitions characterized by populations on instantaneous energy eigenstates are rescaled in time. In the fast-forward scaling theory, there exist phase degrees of freedom. We fix them so that the diagonal part of a total Hamiltonian in the energy-eigenstate basis of a reference Hamiltonian is only given by the reference Hamiltonian in rescaled time and additional driving just contributes to the off-diagonal part. As the result, we find that the additional terms consist of the counterdiabatic term and its similar term. We point out that the latter term reproduces nonadiabatic transitions caused by the reference Hamiltonian in the original time scale. Next, we propose another approach for calculating additional terms. Variety of derivation would enhance its utility. Finally, we discuss the adiabatic limit of reference dynamics. We show that the fast-forward scaling theory for nonadiabatic transitions is asymptotically equivalent to shortcuts to adiabaticity by counterdiabatic driving without introducing any new concept such as the regularization term.

## 2  Fast-forward scaling theory

Here we overview and explain our viewpoint of the fast-forward scaling theory for better understanding of the present results. Note that we adopt a similar notation to Ref. [6] instead of the conventional notation [1, 2].

We introduce reference dynamics $|\Psi_{\mathrm{ref}}(t)\rangle$ governed by a time-dependent Hamiltonian

74 $\hat{H}_{\mathrm{ref}}(t)$. The reference dynamics can be specified by measurement. For example, projection
75 measurement on a certain orthonormal basis $|\sigma\rangle$ gives population of the reference dynamics
76 on this basis

$$|c_\sigma(t)|^2 = |\langle\sigma|\Psi_{\mathrm{ref}}(t)\rangle|^2, \tag{1}$$

77 where $|\Psi_{\mathrm{ref}}(t)\rangle = \sum_\sigma c_\sigma(t)|\sigma\rangle$. The aim of the fast-forward scaling theory is to obtain the
78 same population in different time scale. For this purpose, we introduce rescaled time $s = s(t)$.
79 Then, the aim of the fast-forward scaling theory can be formulated as a problem to find rescaled
80 dynamics (the fast-forward state) $|\Psi_{\mathrm{FF}}(t)\rangle$ and its Hamiltonian $\hat{H}_{\mathrm{FF}}(t)$ satisfying

$$|\langle\sigma|\Psi_{\mathrm{FF}}(t)\rangle|^2 = |\langle\sigma|\Psi_{\mathrm{ref}}(s)\rangle|^2, \tag{2}$$

81 where time scale becomes fast forward for $ds/dt > 1$, slow down for $0 < ds/dt < 1$, a pause
82 for $ds/dt = 0$, and a rewind for $ds/dt < 0$.
83     Since the rescaled dynamics $|\Psi_{\mathrm{FF}}(t)\rangle$ is identical with the reference dynamics at the rescaled
84 time $|\Psi_{\mathrm{ref}}(s)\rangle$ except for phase on the basis $|\sigma\rangle$, it is given by

$$|\Psi_{\mathrm{FF}}(t)\rangle = \hat{U}(t)|\Psi_{\mathrm{ref}}(s)\rangle, \tag{3}$$

85 where $\hat{U}(t)$ is a unitary operator

$$\hat{U}(t) = e^{-i\sum_\sigma f_\sigma(t)|\sigma\rangle\langle\sigma|}, \tag{4}$$

86 with a real number $f_\sigma(t)$. By considering time derivative of rescaled dynamics (3), we can
87 also find its Hamiltonian

$$\hat{H}_{\mathrm{FF}}(t) = \frac{ds}{dt}\hat{U}(t)\hat{H}_{\mathrm{ref}}(s)\hat{U}^\dagger(t) - i\hbar\hat{U}(t)\left(\frac{\partial}{\partial t}\hat{U}^\dagger(t)\right). \tag{5}$$

88     A theoretically trivial example is $\hat{U}(t) = 1\,[f_\sigma(t) = 0]$, which gives $\hat{H}_{\mathrm{FF}}(t) = (ds/dt)\hat{H}_{\mathrm{ref}}(s)$,
89 but it may be experimentally nontrivial. For example, it may require time-dependent mass for
90 quantum particles since the overall amplitude of the reference Hamiltonian must be changed
91 as $ds/dt$ [1]. This example explains why we introduce the unitary operator $\hat{U}(t)$, i.e., it is
92 used to make protocols feasible in experiments.

## 93    3  Nonadiabatic transitions

94 Next we also overview nonadiabatic transitions (for details, see, e.g., Ref. [18]) and shortly
95 mention shortcuts to adiabaticity by counterdiabatic driving [11, 12].
96     In the energy-eigenstate basis, the reference dynamics and its Hamiltonian can be ex-
97 pressed as

$$|\Psi_{\mathrm{ref}}(t)\rangle = \sum_n c_n(t) e^{-\frac{i}{\hbar}\int_0^t dt' E_n(t')}|n(t)\rangle, \tag{6}$$

98 and $\hat{H}_{\mathrm{ref}}(t) = \sum_n E_n(t)|n(t)\rangle\langle n(t)|$. Nonadiabatic transitions are characterized by the abso-
99 lute square of each coefficient of the energy-eigenstate basis [Eq. (1) with $|\sigma\rangle = |n(t)\rangle$], i.e.,
100 $|c_n(t)|^2 = |\langle n(t)|\Psi_{\mathrm{ref}}(t)\rangle|^2$. The Schrödinger equation gives its time evolution

$$i\hbar\frac{\partial}{\partial t}c_n(t) + i\hbar\sum_m \langle n(t)|\left(\frac{\partial}{\partial t}|m(t)\rangle\right)c_m(t) = 0. \tag{7}$$

101 Here, the off-diagonal part of the second term causes transitions between different levels, i.e.,
102 it describes nonadiabatic transitions. The operator form of the second term is given by

$$\hat{H}_{\mathrm{cd}}(t) = i\hbar\sum_{\substack{n,m \\ (n\neq m)}} |n(t)\rangle\langle n(t)|\left(\frac{\partial}{\partial t}|m(t)\rangle\right)\langle m(t)|, \tag{8}$$

103 which is known as the adiabatic gauge potential [18] or the counterdiabatic term [11,12]. In
104 counterdiabatic driving, we apply this term to the reference Hamiltonian, and then nonadia-
105 batic transitions are canceled out and the solution of the Schrödinger equation becomes the
106 adiabatic state [11,12].

## 4  Time rescaling of nonadiabatic transitions

108 Now we discuss time rescaling of nonadiabatic transitions. The condition for the rescaled
109 dynamics (2) is rewritten as

$$|\langle n(s)|\Psi_{\text{FF}}(t)\rangle|^2 = |\langle n(s)|\Psi_{\text{ref}}(s)\rangle|^2. \tag{9}$$

110 Note that the energy-eigenstate basis in the left-hand side of this equation is that in the rescaled
111 time $s$, whereas the rescaled dynamics is in the usual time scale $t$. Such dynamics is given by
112 Eq. (3) and Eq. (4) with $|\sigma\rangle = |n(s)\rangle$.
113     Then we discuss the rescaled Hamiltonian (5). The first term in the rescaled Hamiltonian
114 (5) is simply given by $(ds/dt)\hat{U}(t)\hat{H}_{\text{ref}}(s)\hat{U}^\dagger(t) = (ds/dt)\hat{H}_{\text{ref}}(s)$, i.e., it only gives the diag-
115 onal term in the energy-eigenstate basis. In addition, the diagonal term in the second term
116 of the rescaled Hamiltonian (5) is given by $\hbar \sum_n (df_n/dt)|n(s)\rangle\langle n(s)|$. Therefore, by setting
117 $\hbar(df_n/dt) = (1 - ds/dt)E_n(s)$, the diagonal part of the total rescaled Hamiltonian (5) be-
118 comes $\hat{H}_{\text{ref}}(s)$. For this phase $f_n(t)$, we can also calculate off-diagonal terms, and finally we
119 find that the total rescaled Hamiltonian (5) is given by

$$\hat{H}_{\text{FF}}(t) = \hat{H}_{\text{ref}}(s) + \frac{ds}{dt}[\hat{H}_{\text{cd}}(s) + \hat{H}_{\text{nad}}(t)], \tag{10}$$

120 where the second term $(ds/dt)\hat{H}_{\text{cd}}(s)$ is the counterdiabatic term (8) for the reference Hamil-
121 tonian in the rescaled time $\hat{H}_{\text{ref}}(s)$, and the third term is our finding given by

$$\hat{H}_{\text{nad}}(t) = -i\hbar \sum_{\substack{n,m \\ (n \neq m)}} e^{-i[f_n(t)-f_m(t)]}|n(s)\rangle\langle n(s)|\left(\frac{\partial}{\partial s}|m(s)\rangle\right)\langle m(s)|. \tag{11}$$

122 Remarkably, the matrix element of the third term (11) is given by

$$\langle n(s)|\hat{H}_{\text{nad}}(t)|m(s)\rangle = -e^{-i[f_n(t)-f_m(t)]}\langle n(s)|\hat{H}_{\text{cd}}(s)|m(s)\rangle, \tag{12}$$

123 that is, the third term (11) is the opposite sign of the counterdiabatic term (8) with the phase
124 factor associated with the rescaling rate $ds/dt$. Note that the third term (11) completely
125 cancels out the second term (8) when the rescaled time $s$ is equal to the time $t$, and thus the
126 rescaled Hamiltonian (10) recovers the reference Hamiltonian $\hat{H}_{\text{ref}}(t)$. According to the theory
127 of counterdiabatic driving [11,12], the second term $(ds/dt)\hat{H}_{\text{cd}}(s)$ cancels out diabatic changes
128 caused by the first term $\hat{H}_{\text{ref}}(s)$, and thus we can conclude that the third term $(ds/dt)\hat{H}_{\text{nad}}(t)$
129 reproduces nonadiatic transitions caused by the reference Hamiltonian in the original time
130 scale $\hat{H}_{\text{ref}}(t)$.
131     Finally, we propose another way for constructing the third term (11). As in the case of
132 counterdiabatic driving, it is not always easy to construct additional driving from its mathe-
133 matical expression. Indeed, we have to find explicit expression of operators from off-diagonal
134 elements $|n(s)\rangle\langle m(s)|$. The key idea of our proposal is use of the following formula [19]

$$e^{-\hat{O}(t)}\frac{\partial}{\partial t}e^{\hat{O}(t)} = \sum_{k=0}^{\infty}\frac{(-1)^k}{(k+1)!}(\text{ad}_{\hat{O}(t)})^k\frac{\partial}{\partial t}\hat{O}(t), \tag{13}$$

for the second term of Eq. (5), where $\hat{O}(t) = i\sum_n f_n(t)|n(s)\rangle\langle n(s)|$ in the present paper and $\mathrm{ad}_{\hat{O}}\bullet = [\hat{O}, \bullet]$ is the adjoint action, i.e., $(\mathrm{ad}_{\hat{O}})^k \bullet = [\hat{O}, [\hat{O}, \dots [\hat{O}, \bullet]\dots]]$ is the $k$th nested commutator. That is, once we find the explicit expression of $\hat{O}(t)$, which can be calculated by using diagonal elements $|n(s)\rangle\langle n(s)|$, we can construct the second term of Eq. (5) by calculating the nested commutators. Notably, the counterdiabatic term (8) can also be calculated by using the nested commutators of the reference Hamiltonian [20–22], and thus we can extract the third term (11) from the results. Because difficulty in finding operator forms from off-diagonal elements $|n(s)\rangle\langle m(s)|$ and diagonal elements $|n(s)\rangle\langle n(s)|$ could differ depending on given systems, this formula has potential usefulness for constructing the additional term (11). Moreover, for a $D$-dimensional quantum system, the number of elements is $D(D-1)/2$ for off-diagonal elements (and their Hermitian conjugates), but it is $D$ for diagonal elements.

# 5  Adiabatic limit

Now we discuss asymptotic behavior of our results in the adiabatic limit of reference dynamics. In the conventional formalism of the fast-forward scaling theory for adiabatic time evolution [2, 10], we have to introduce the "regularization term" for its justification since the adiabatic state is not the solution of the Schrödinger equation under a given reference Hamiltonian. Later it was pointed out that this regularization term is the counterdiabatic term or the single-eigenstate counterdiabatic term [16]. Here we propose another interpretation without introducing such an addition concept. Note that for simplicity we set $\hbar = 1$ and assume that all time scale and energy scale are dimensionless (we can easily recover dimension by multiplying $\hbar$ in an appropriate way).

Adiabatic time evolution is realized under slow change of parameters. Roughly speaking, the operation time should be larger than the inverse square of the minimum energy gap, $T_{\mathrm{ad}} \gg (\Delta E_{\min})^{-2}$, where $T_{\mathrm{ad}}$ is the adiabatic time scale and $\Delta E_{\min}$ is the minimum energy gap. We assume that the operation time of the reference dynamics $T_{\mathrm{ref}}$ is long enough compared with this adiabatic time scale, $T_{\mathrm{ref}} \gtrsim T_{\mathrm{ad}}$. By using the fast-forward scaling theory, we can realize this adiabatic time evolution within shorter time, say the fast-forwarded time scale $T_{\mathrm{FF}}$, where $T_{\mathrm{FF}} \ll T_{\mathrm{ad}} \lesssim T_{\mathrm{ref}}$. Then, for example, the rescaled time $s$ can be expressed as $s(t) = (T_{\mathrm{ref}}/T_{\mathrm{FF}})t$. Since $ds/dt = T_{\mathrm{ref}}/T_{\mathrm{FF}} \gg 1$, the leading term in the phase factor of the third term $e^{-i[f_n(t)-f_m(t)]}$ is given by $|\int_0^t dt'(T_{\mathrm{ref}}/T_{\mathrm{FF}})[E_m(s(t'))-E_n(s(t'))]| \geq |T_{\mathrm{ref}}(t/T_{\mathrm{FF}})\Delta E_{\min}|$. Since $t/T_{\mathrm{FF}}$ is a linear sweep from 0 to 1, the leading value of the phase is determined by the relation between the reference time scale $T_{\mathrm{ref}}$ and the minimum energy gap $\Delta E_{\min}$. In the adiabatic limit, $T_{\mathrm{ref}} \gtrsim T_{\mathrm{ad}} \gg (\Delta E_{\min})^{-2}$, this term gives high-frequency oscillation, and thus the third term in the rescaled Hamiltonian (11) effectively vanishes. As the result, the rescaled Hamiltonian (10) becomes the summation of the reference Hamiltonian in the rescaled time scale and its counterdiabatic Hamiltonian, i.e., $\hat{H}_{\mathrm{FF}}(t) \approx \hat{H}_{\mathrm{ref}}(s) + (ds/dt)\hat{H}_{\mathrm{cd}}(s)$. In conclusion, we find that the fast-forward scaling theory for adiabatic time evolution is asymptotically equivalent to shortcuts to adiabaticity by counterdiabatic driving.

# 6  Example

Finally, we consider an example. As the reference Hamiltonian, we consider a two-level system

$$\hat{H}_{\mathrm{ref}}(t) = -h^x(t)\hat{X} - h^z(t)\hat{Z}, \tag{14}$$

where $h^x(t)$ and $h^z(t)$ are a time-dependent transverse field and a time-dependent longitudinal field. Here we express the Pauli matrices as $\{\hat{X}, \hat{Y}, \hat{Z}\}$. The eigenenergies and their eigenstates

are given by

$$
\begin{cases}
E_\pm(t) = \pm\sqrt{h^{x2}(t) + h^{z2}(t)}, \\
|+(t)\rangle = \begin{pmatrix} -\sin\theta(t) \\ \cos\theta(t) \end{pmatrix}, \quad |-(t)\rangle = \begin{pmatrix} \cos\theta(t) \\ \sin\theta(t) \end{pmatrix}
\end{cases}
\tag{15}
$$

where $\theta(t)$ satisfies

$$
\begin{cases}
\sin 2\theta(t) = \dfrac{h^x(t)}{\sqrt{h^{x2}(t) + h^{z2}(t)}}, \\
\cos 2\theta(t) = \dfrac{h^z(t)}{\sqrt{h^{x2}(t) + h^{z2}(t)}}.
\end{cases}
\tag{16}
$$

First, we construct the additional term by using the formula (11). The operator form of the off-diagonal element is given by $|+(s)\rangle\langle-(s)| = (1/2)\cos 2\theta(s)\hat{X} - (i/2)\hat{Y} - (1/2)\sin 2\theta(s)\hat{Z}$, and its coefficient is given by $\langle+(s)|(\partial/\partial s)|-(s)\rangle = \partial\theta(s)/\partial s$. The phase factor is given by $e^{-i[f_+(t)-f_-(t)]} = e^{-2if_+(t)}$, where $f_+(t) = \int_0^t dt'(1 - ds/dt')E_+(s)$. Therefore, we find that the additional term (11) is given by

$$
\hat{H}_{\mathrm{nad}}(t) = -\frac{\partial\theta(s)}{\partial s}\sin 2f_+(t)[\cos 2\theta(s)\hat{X} - \sin 2\theta(s)\hat{Z}] - \frac{\partial\theta(s)}{\partial s}\cos 2f_+(t)\hat{Y}.
\tag{17}
$$

Note that we can also construct the counterdiabatic term from Eq. (8) by using the above equations and it is given by

$$
\hat{H}_{\mathrm{cd}}(s) = \frac{\partial\theta(s)}{\partial s}\hat{Y}.
\tag{18}
$$

Next, we construct the additional term by using the formula (13). The operator forms of the diagonal elements are given by $|\pm(s)\rangle\langle\pm(s)| = (1/2)\hat{1}\mp(1/2)\sin 2\theta(s)\hat{X}\mp(1/2)\cos 2\theta(s)\hat{Z}$, where the double sign corresponds and $\hat{1}$ is the identity operator, and thus we obtain $\hat{O}(t) = -if_+(t)[\sin 2\theta(s)\hat{X} + \cos 2\theta(s)\hat{Z}]$. Note that $(\partial/\partial t)\hat{O}(t) = i(1 - ds/dt)\hat{H}_{\mathrm{ref}}(s) - 2i(ds/dt)$ $(\partial\theta(s)/ds)f_+(t)[\cos 2\theta(s)\hat{X} - \sin 2\theta(s)\hat{Z}]$ and $[\hat{O}(t), \hat{H}_{\mathrm{ref}}(s)] = 0$. Owing to the algebraic structure of the nested commutators, we obtain

$$
(\mathrm{ad}_{\hat{O}(t)})^k\frac{\partial}{\partial t}\hat{O}(t) = \frac{ds}{dt}\frac{\partial\theta(s)}{\partial s}(-i)^{k+1}[2f_+(t)]^{k+1}\hat{W}_k, \quad \text{for} \quad k > 0,
\tag{19}
$$

where $\hat{W}_k = i\hat{Y}$ for odd $k$ and $\hat{W}_k = \cos 2\theta(s)\hat{X} - \sin 2\theta(s)\hat{Z}$ for even $k$. By substituting this result for Eq. (5), we obtain Eq. (10) with Eqs. (14), (17), and (18). As mentioned in the general discussion, the counterdiabatic term (18) can be specified by using the nested commutators of the reference Hamiltonian (14), and thus we can extract Eq. (17).

Finally, we consider the adiabatic limit of the reference dynamics. Here we again assume the linear rescaling $s(t) = (T_{\mathrm{ref}}/T_{\mathrm{FF}})t$ and fast-forwarding $T_{\mathrm{ref}}/T_{\mathrm{FF}} \gg 1$. In the present example, the third term (17) oscillates with phase $2f_+(t)$. The leading term of this phase is given by $|2f_+(t)| \approx |\int_0^t dt'(T_{\mathrm{ref}}/T_{\mathrm{FF}})\Delta E(s)| \geq |T_{\mathrm{ref}}(t/T_{\mathrm{FF}})\Delta E_{\mathrm{min}}|$, where $\Delta E(s) = E_+(s) - E_-(s)$ is an energy gap. As mentioned in the general discussion, we find that it causes fast oscillation in the adiabatic limit, $T_{\mathrm{ref}} \gtrsim T_{\mathrm{ad}} \gg (\Delta E_{\mathrm{min}})^{-2}$, and thus the total rescaled Hamiltonian is given by $\hat{H}_{\mathrm{FF}}(t) \approx \hat{H}_{\mathrm{ref}}(s) + (ds/dt)\hat{H}_{\mathrm{cd}}(s)$.

# 7 Conclusion

In this paper, we discussed time rescaling of nonadiabatic transitions by using the fast-forward scaling theory. We found that the additional terms consist of the counterdiabatic term (8) and

its similar term (11). We pointed out that the latter term (11) reproduces nonadiabatic transitions caused by the reference Hamiltonian in the original time scale. Moreover, we showed that the third term (11) effectively vanishes in the adiabatic limit due to fast oscillation. As the result, the fast-forward scaling theory for nonadiabatic transitions asymptotically reproduces counterdiabatic driving of shortcuts to adiabaticity.

We proposed two ways for calculating the additional term, i.e., Eq. (11) and Eq. (13). Although these formulae use different elements in the energy-eigenstate basis, the knowledge of the energy eigenstates of the reference Hamiltonian is required. It is the important future work to find methods for constructing the additional term without the knowledge of the energy eigenstates as in the case of counterdiabatic driving of shortcuts to adiabaticity [20–22].

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
