# Peer review of "Time rescaling of nonadiabatic transitions"

_SciPost Physics_

## Round 1 · Referee Report · Anonymous (Referee 1) · 2023-3-7

Strengths

Provides a direct bridge between two popular theoretical approaches to quantum control. Presentation is to the point.

Weaknesses

None that I was able to identify.

Report

In this paper, the author studies the relation between the so-called fast-forward driving scheme and the shortcut to adiabaticity by counterdiabatic driving, as applied to the time-dependent evolution of a quantum system.
The former approach considers how to modify the Hamiltonian to recover the result of a slow parameter quench by driving that is performed at shorter time scales. The latter considers how to modify the Hamiltonian to recover the adiabatic limit of a very slow evolution along the ground states (eigenstates) manifold with parameter driving that is performed at a finite time. As can be naturally expected, both approaches are closely related, which gets nicely and pedagogically elucidated in this article.

The article bridges two popular theoretical approaches to quantum control that appear in the literature. The text is well written, both concise and self-standing. The citation list looks appropriate. I believe the article satisfies the expected acceptance criteria. As such, I can only recommend the publication of this article in SciPost Physics in its current form.

---

## Round 1 · Referee Report · Anonymous (Referee 2) · 2023-3-28

Strengths

The topic is very interesting, presented in a concise and self-standing way.

Weaknesses

None

Report

The author applies the fast-forward scaling theory to a system by rescaling its non-adiabatic transitions. They find that the "fast-forwarded" Hamiltonian consists of the original Hamiltonian plus two additional terms, namely the counterdiabatic term (as in shortcuts to adiabaticity) and another similar term. They also explicit two ways in which this new term can be calculated. Furthermore they show how the theory of shortcuts to adiabaticity can be recovered from fast-forward scaling theory of non-adiabatic transitions in the adiabatic limit of the reference Hamiltonian.

Their results allow to connect the fast-forward scaling theory with the theory of shortcuts to adiabaticity and is thus very interesting and I suggest the publication in SciPost Physics.

Requested changes

The paper is well written, my only suggestion, at the discretion of the author, is to improve the last paragraph of the introduction (in which they summarize the points discussed in the paper) in order to make it easier to understand. To me it became clear only after reading the paper.

---

## Editorial Decision

resubmitted